# In vitro cytotoxicity (irritant potency) of toothpaste ingredients

**Shaira R. Kasi[1]☯, Sanne Roffel[2]☯, Mutlu Özcan[3]☯, Susan Gibbs[4,5,6]☯, Albert J. Feilzer[1]☯\***

**1** Department of Dental Materials Science, Academic Center for Dentistry (ACTA), University of Amsterdam, Amsterdam, The Netherlands, **2** Department of Oral Cell Biology, University of Amsterdam, Academic Center for Dentistry (ACTA), Amsterdam, The Netherlands, **3** Clinic for Masticatory Disorders and Dental Biomaterials, University of Zurich, Center for Dental Medicine, Zurich, Switzerland, **4** Department of Oral Cell Biology, Amsterdam, VU University Medical Centre, Academic Centre for Dentistry (ACTA), Amsterdam, The Netherlands, **5** Molecular Cell Biology & Immunology, Amsterdam UMC, Amsterdam, The Netherlands, **6** Amsterdam Institute for Immunity and Infectious Diseases, Amsterdam UMC, Amsterdam, The Netherlands

☯ These authors contributed equally to this work.
\* a.feilzer@acta.nl

**Data Availability Statement:** All relevant data are within the manuscript and its Supporting Information files.

**Funding:** The author(s) received no specific funding for this work.

## Abstract

### Purpose

This study aimed to determine the cytotoxicity (irritant potency) of toothpaste ingredients, of which some had known to have sensitizing properties.

### Materials

From the wide variety of toothpaste ingredients, Xylitol, Propylene glycol (PEG), Sodium metaphosphate (SMP), Lemon, Peppermint, Fluoride, Cinnamon, and Triclosan and Sodium dodecyl sulphate (SDS) have been selected for evaluation of their cytotoxic properties.

### Methods

Reconstructed human gingiva (RHG) were topically exposed to toothpaste ingredients at different concentrations. The compound concentration resulting in 50% cell death (EC50) and 10% cell death (EC10) was determined by the MTT assay. Detrimental effects in tissue histology were observed by hematoxylin & eosin staining of tissue sections followed by microscopy.

### Results

While Xylitol, PEG, and SMP did not appear to affect cell viability or tissue histology, the concentrations of Lemon, Peppermint, Cinnamon and SDS present in toothpastes exceeded the EC50 value and resulted in clear detrimental effects in tissue histology, indicating that they could harm the oral mucosa. Triclosan and Fluoride concentrations in the tested toothpastes exceeded the EC10 value but remained below the EC50 value with no clear detrimental effects in tissue histology.

**Competing interests:** The authors have declared that no competing interests exist.

## Clinical significance

Manufacturers are encouraged to comply with higher standards of quality and safety for toothpaste.

## Introduction

For over a century, brushing teeth with toothpaste has been the standard for maintaining oral hygiene. The addition of fluoride in the 70-ties is probably responsible for the huge reduction in caries prevalence over the past 50 years. Since then, toothpastes have been developed with a variety of claims regarding their effect and have become complex healthcare products. Today's toothpaste often contains more than 20 ingredients on average, out of around 185 different ingredients. In general, the components that makeup toothpaste can be divided into two categories. In addition to 'co-formulations' that form the basis for making a paste, 'active ingredients' or 'therapeutic excipients' are added to promote the therapeutic effects of the toothpaste.

There are five known categories for co-formulations as well as viscosity modifiers to ensure that the toothpaste has and retains a paste-like consistency. While humidifiers maintain moisture and prevent toothpaste from drying out due to exposure to the air, flavors, sweeteners, or fragrances are also used to make the toothpaste more attractive and tolerable. While coloring dyes are used for attractiveness, the preservatives prevent the growth of microorganisms in toothpaste.

There are 13 categories for known active ingredients each with their own intended purpose as claimed by manufacturers [1]. Active ingredients include anti-caries (e.g. fluoride), anti-plaque (e.g. triclosan), anti-gingivitis, anti-calculus, anti-xerostomia, non- erosive agents along with fragrances against bad mouth odor, remineralization agents to reduce tooth sensitivity and bleaching or whitening agents (e.g. sodium metaphosphate) to promote teeth whitening or to remove surface discolorations. Other agents with the postulated healing effect for the soft tissues are aloe vera, coconut leaves, and alike.

Detergents or surfactants (e.g. Sodium dodecyl sulphate) on the other hand give a soapy effect so that the brushed particles do not stick to the tooth tissue and can be washed away. Moreover, detergents or surfactants have also a foaming effect which suggests to the user that the toothpaste is active. And finally, abrasives are added to remove the discolorations such as stains on the tooth surface due to smoking.

Despite its good qualities, some people do not tolerate toothpaste well. Nearly all toothpastes contain components that people may react with a cytotoxic and/or allergic reaction. For cytotoxic reactions, the concentrations must pass a relatively higher threshold concentration value than for allergic reactions which can be extremely low. Previously, the local lymph node assay was used to test the sensitizing potential of a cosmetic chemical and the rabbit eye Draize test was used to test the irritant potential. Both of these animal-based tests have now been replaced with EURL-ECVAM validated in vitro alternative tests [2–5]. In order to predict irritation and corrosion properties a reconstructed human epidermis model is used. This assay involves the topical application of the test substance to the stratum corneum of the epidermis followed by cytotoxicity assessment by measuring a decrease in metabolic activity with the MTT assay [2]. The MTT assay is a colorimetric assay that assesses cell metabolic activity, which is an indicator of cell viability and proliferation. Therefore, a decrease in metabolic activity correlates to cytotoxicity. In the past, we have adapted this assay to address contact sensitizer and irritant potency by determining the EC50 value for a chemical (the chemical

concentration that reduces cell viability by 50%) [6]. Chemicals correlating to a low EC50 value are more cytotoxic than chemicals corresponding to a high EC50 value.

Toothpastes are orally applied and oral mucosa is considerably more permeable to substances than the skin. Consequently, it is reasonable to assume that toxic agents applied orally may impact cell viability at lower concentrations compared to their application on the skin. Therefore, in this present study, the Reconstructed Human Epidermis potency assay has been further adapted for testing toothpaste ingredients in a bilayered reconstructed Human Gingiva (RHG) model consisting of a differentiated and stratified gingival epithelium on a gingival fibroblast populated lamina propria hydrogel [7]. In addition to determining the EC50 value, the EC10 value (chemical concentration resulting in 10% cell death, 90% cell viability) was determined as this represents the concentration at which an observable effect initiates. The difference between the two values indicates the risk or the susceptibility to toxic effects when there are slight alterations in concentration. The present study investigated the impact of some of the soluble compounds present in commercially available toothpastes on the viability of RHG. The highest compound concentration tested was 200 mg/ml as this exceeds concentrations used in toothpaste products for all tested compounds. In addition to determining EC50 and EC10 values, the histology of RHG tissue sections was assessed for any detrimental effects caused by topical exposure to the compounds.

## Material and methods

### Material selection

In a former internal, unpublished, study at ACTA on the composition of toothpastes, the ingredients of 180 different commercially available toothpastes were classified. This resulted in 186 different ingredients ranging from water to fluorides. For all ingredients, we then searched Pubmed for scientific evidence on whether an ingredient was known to cause an irritant reaction. This refined search identified nine toothpaste ingredients with known irritant properties (Table 1). Below, these are summarized based on their composition: Triclosan: A non-ionic broad-spectrum antibacterial agent added to many consumer products and is intended to reduce bacterial contamination. Triclosan was chosen for this study based on the fact that it was withdrawn from the market by many toothpaste manufacturers due to its toxicity. Triclosan is one of the most effective antimicrobial agents in toothpaste because of its good antimicrobial activity [8]. It is also known that Triclosan can cause some untoward reactions to the oral mucosa [4,9,10]. The estimated weight in toothpaste for Triclosan is 0.1–0.3 w% [11].

*Sodium dodecyl* sulphate (SDS), a detergent also known as Sodium lauryl sulphate, is an anionic surfactant that is mainly chosen as an ingredient for toothpaste because of its cleansing

**Table 1. Toothpaste ingredients that were tested.**

| Toothpaste Ingredient | Manufacturer | Batch | Test vehicle | Hazard label |
|---|---|---|---|---|
| Triclosan | EMD Millipore | 647950 | AOO | Toxic, irritant |
| Sodium dodecyl sulphate (SDS) | Sigma-Aldrich | L3771 | Water | Irritant |
| Propylene glycol (PEG) | Sigma-Aldrich | W294004 | Water | Sensitizer, Irritant |
| Sodium metaphosphate (SMP) | Sigma-Aldrich | T5508 | Water | Moderately toxic if ingested in large amounts. |
| Fluoride (natrium) | Sigma-Aldrich | 201154 | Water | Toxic |
| Peppermint Oil | Sigma-Aldrich | W284815 | AOO | Irritant |
| Cinnamaldehyde | Sigma-Aldrich | W228613 | AOO | Sensitiser, irritant |
| Lemon (® (+) Limonene) | Sigma-Aldrich | 183164 | AOO | Sensitiser, irritant |
| Xylitol | Sigma-Aldrich | X3375 | Water | May cause gastrointestinal disturbances |

and foaming effect. Despite these desired effects, it is known that SDS can irritate the mucous membranes even in low quantities. It is also often claimed that SDS-free toothpaste helps patients to prevent the development of aphthous lesions as well as to allow the wounds to heal faster. Recently an article was published that suggests the cleansing effect of SDS on microorganisms dislodged from the biofilm by brushing is not relevant [12]. The estimated amount in weight in toothpaste for SDS is 1.2 w% [13].

*Propylene glycol (PEG)* is a viscous liquid that is often used as a humectant in toothpaste. However, PEG is also a known cause of irritant and allergic contact dermatitis [14]. The estimated amount in weight in toothpaste for PEG is 0.5–2 w%.

*Sodium metaphosphate*: An inorganic salt that is often used as an abrasive in toothpaste. Sodium metaphosphate (SMP) is the general term for various polyphosphates. In certain concentrations, it may cause irritations and other side effects [15]. The estimated amount in weight in toothpaste for SMP is 8 w%.

*Fluoride* is used as an anti-caries and anti-plaque agent in toothpaste. The beneficial effects of fluoride on human oral health are well studied and a small amount of fluoride delivered to the oral cavity decreases the prevalence of dental decay and results in stronger teeth [16]. Click or tap here to enter text [16,17]. However, ingestion of fluoride more than the recommended limit leads to toxicity and adverse effects. Children are known to sometimes ingest more than 30% of their toothpaste [17,18]. Fluoride can also damage the intestinal structure, disturb the intestinal micro-ecology, and cause microflora disorder in mice [18,19]. The estimated amount in weight in toothpaste for Fluoride is 0.26 w% [20].

*Mint*, *Cinnamon*, and *Lemon* are used as flavors in toothpaste and are well-known sensitizers [21–23]. The estimated amount in weight in toothpaste for mint is 0.5–0.9 w%, for Cinnamon, is 1w %, and for Lemon is 5 w% [24–26]. (Magnusson and Wilkinson 1975; Nair 2001; Xie et al. 2010)

*Xylitol* is known as a sweetener in toothpaste. The estimated amount in weight in toothpaste for Xylitol is 10 w% [7] (Lif Holgerson et al. 2005).

## Reconstructed Human Gingiva (RHG)

Reconstructed Human Gingiva (RHG) were constructed essentially as previously described. Immortalized human gingiva cell lines (fibroblasts and keratinocytes) were used to construct RHG: fibroblast (Fib-TERT, T0026, ABM, Richmond, BC, Canada) and keratinocyte (KC-TERT, OKG4/bmi1/TERT, Rheinwald laboratory, Boston, MA, USA). The lamina propria hydrogel compartment was constructed by mixing rat-tail collagen with 1:1 fibrinogen (Diagnostica Stago S.A.S., Asnieres sur Seine, France), Hank's Balanced Salt Solution (HBSS; Gibco, Grand Island, USA) (HBSS was diluted 8-fold) and fibroblasts ($3.25 \times 10^4$ cells/gel) [7]. (Each hydrogel contained 0.5 U/ml thrombin (Merck KGaA, Darmstadt, Germany) to allow for fibrin formation. The fibroblast-populated hydrogels were cultured in a twelve-well transwell insert with 0.4 μm pores (ThinCerts™, Greiner Bio/One, Alphen aan den Rijn, Netherlands) and incubated overnight submerged in fibroblast medium (DMEM, 5% Fetal Clone III (GE Healthcare, Chicago, USA), 1% penicillin-streptomycin (Gibco)). Hereafter, keratinocytes ($1.25 \times 10^5$ cells/transwell) were pipetted on top of the fibroblast- populated hydrogel. After a further three days submerged culture in KC medium (DMEM/Ham's F12 (3/1) (Gibco), supplemented with 5% Fetal Clone III (GE), 1% penicillin–streptomycin (Gibco), 1 μM hydrocortisone (Sigma-Aldrich, St. Louis, USA), 0.1 μM insulin (Sigma-Aldrich), 1 μM isoproterenol (Sigma-Aldrich), and 2 ng/mL epidermal growth factor (EGF, Sigma-Aldrich), RHG was lifted at the air-liquid interface and further cultured for ten days in differentiation medium (DMEM/Ham's F12 (3/1), supplemented with 1% Fetal Clone III, 1% penicillin–streptomycin,

2 μM hydrocortisone, 0.1 μM insulin, 1 μM isoproterenol, 10 μM carnitine (Sigma-Aldrich), 10 mM L-serine (Sigma-Aldrich), 0.4 mmol l-ascorbic acid (Sigma-Aldrich), and 2 ng/mL EGF) in which time a differentiated, stratified epithelium developed on the fibroblast-populated hydrogels. Culture medium was refreshed twice in this period.

Before the exposure, cultures were incubated 24 h in the above-mentioned differentiation medium but in the absence of hydrocortisone. Hereafter, cultures were exposed to compounds.

All experiments were conducted in two independent experiments, each using a different batch of RHG which had been grown separately at different times.

## Compound exposure

The solubility of the compounds was tested by dissolving the compounds in two vehicles: water and acetone (Sigma-Aldrich)-olive oil (Sigma-Aldrich) mix (AOO, 4:1) (Table 1). The vehicle with the highest dissolving capacity was chosen. AOO was chosen as the vehicle for Triclosan, Mint, Cinnamon, and Lemmon. Water was the chosen vehicle for the other compounds. For the exposure, gauze discs (Sefar Nitex, Heiden, Switzerland) of 12 mm diameter were impregnated with the compounds and applied topically to the RHG (Fig 1). Compound exposures were performed using a single dose-response with 2-fold serial dilutions starting from 200 mg/ml or the maximum soluble concentration according to our standard operating procedure for 24 hours. (Teunis et al. 2014) After compound exposure, gauze discs were

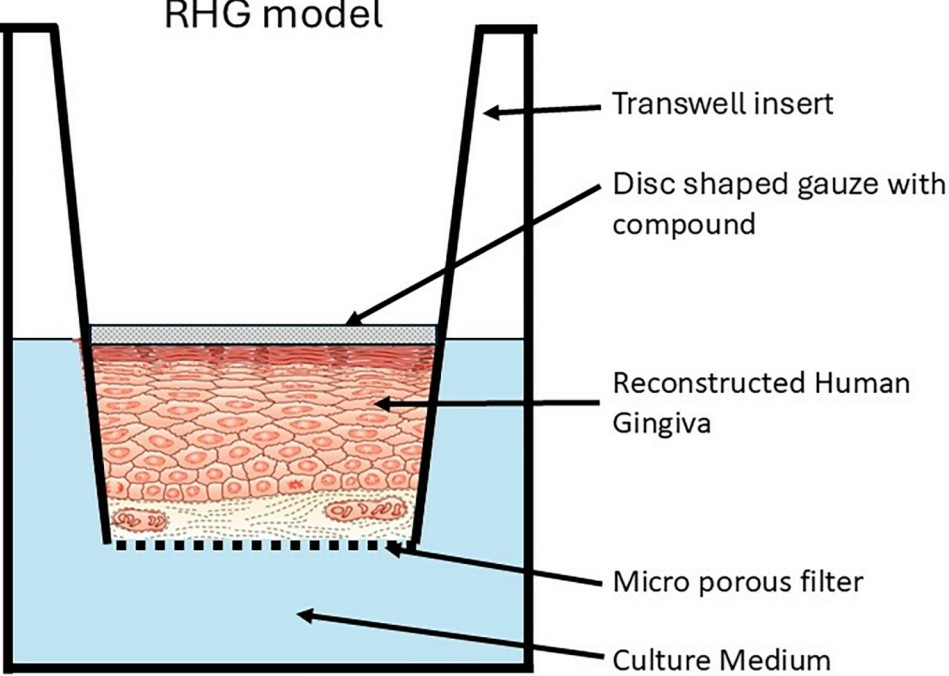

**Fig 1. Graphical representation of the RHG culture.** Immortalized human gingiva cell lines (fibroblasts and keratinocytes) were used to construct RHG: Fibroblast-populated hydrogels were cultured in a twelve-well transwell insert with 0.4 μm pore and incubated overnight submerged in fibroblast medium. Hereafter, keratinocytes were pipetted on top of the fibroblast- populated hydrogel. After a further three days submerged culture in KC medium RHG was lifted at the air-liquid interface and further cultured for ten days in differentiation medium in which time a differentiated, stratified epithelium developed on the fibroblast-populated hydrogels. Before the exposure, cultures were incubated 24 h in the above-mentioned differentiation medium but in the absence of hydrocortisone. Hereafter, cultures were exposed to compounds dissolved in a disc shaped gauze.

removed and RHGs were rinsed in PBS. Hereafter, they were cut in half with one half for determination of metabolic activity (MTT assay) and the other half for histology.

### RHG histology

One half of each RHG's was fixed in 4% paraformaldehyde (Klinipath/VWR, Amsterdam, Netherlands) and processed for conventional paraffin embedment. Embedded tissues were cut into 5 μm sections for staining with hematoxylin and eosin (H&E). Images were taken using a Nikon Eclipse 80i microscope with NIS-Elements software (Nikon Instruments Europe B.V.).

### MTT assay

The MTT assay (Sigma–Aldrich) was used to determine the cytotoxicity of the tested compounds as described previously [27]. After removal of the gauze with compounds, RHG's were rinsed in PBS. RHG's were cut in half and half of the RHG was placed in a 24 wells plate with MTT solution (500μl of a 2 mg/mL Thiazolyl blue tetrazolium bromide solution in PBS) and incubated for 2 hr at 37˚C. After that, the RHG's were transferred to a new 24-well plate with 500 μl acidified Isopropanol (Sigma-Aldrich). The plates were incubated at 4˚C overnight and the absorbance was measured at 570 nm and expressed as a percentage relative to the absorbance value of the control vehicle.

To determine whether compounds interfered with the readout of the MTT assay which would indicate that they fall outside of the applicability domain of the assay and would need to be excluded, the highest soluble compound concentration was tested in the absence of RHG. If a colour change was observed then the compound was excluded. None of the compounds were excluded.

The data from the MTT assay are per RHG exposed to a test material compared to that of unexposed RHG of the same RHG batch as well as to the values obtained from exposure to the test vehicle only. The data are shown normalized to the results of the exposure to the test vehicle per culture.

### Determination of irritant potency: EC50 and EC10 values

Irritant potency assessment:

i.  EC50 and EC10 values are the compound concentrations required to reduce metabolic activity (corresponding to cell viability) to 50% or 10% respectively of the value obtained by the vehicle (water or AOO). Values were obtained by 4 point logistic analysis based on changes in metabolic activity (MTT). The web-based EC50 Calculator of AAT Bioquest [28] was used.

ii.  The difference between the EC50 and EC10 values indicates the risk or the susceptibility to toxic effects when there are slight alterations in concentration.

Results are obtained from two independent experiments each with a separate batch of RHG grown at a different time. Data is reported as average and standard deviation of the two obtained data values obtained for each chemical relative to its vehicle.

## Results

To get an indication as to whether the compounds were cytotoxic to RHG, each compound was first tested at 200 mg/ml or at the concentration correlating to their maximum solubility in either AOO or water. 200 mg/ml is a compound concentration that exceeds that used in toothpaste for all tested compounds (Table 2). Topical application of Xylitol, PEG, and SMP to

**Table 2. Results of RHG irritant potency assay.**

| Concentration mg/ml | Triclosan | SDS | PEG | SMP | Fluoride | Cinnamon | Peppermint | Lemon | Xylitol |
|---|---|---|---|---|---|---|---|---|---|
| | Normalized Viability (%) | | | | | | | | |
| *vehicle normalized (0)* | *100 (5,5)* | *100 (4.0)* | *100 (3.1)* | *100 (3.1)* | *100 (7.6)* | *100 (7.9)* | *100 (5.7)* | *100 (7.9)* | *100 (0.4)* |
| **0.1** | 76.2 (11.4) | 82.0 (0.1) | 95.4 (3.3) | 86.7 (0.8) | 102.4 (19.3) | 109.8 (8.1) | 96.8 (2.3) | 108.3 (9.5) | - |
| **0.2** | 74.5 (0.7) | 69.9 (38.0) | 101.0 (0.0) | 98.1 (1.9) | 97.6 (12.2) | 107. (7.6) | 97.2 (3.2) | 111.1 (5.6) | - |
| **0.39** | 69.0 (17.6) | 66.6 (46.6) | 94.9 (0.6) | 100.8 (4.7) | 103.2 (14.7) | 95.5 (1.7) | 97.1 (0.2) | 109.9 (1.7) | - |
| **0.78** | 65.4 (21.8) | 29.1 (32.7) | 90.0 (5.8) | 94.3 (10.9) | 103.7 (24.7) | 95.0 (0.9) | 81.8 (0.2) | 107.4 (3.0) | - |
| **1.56** | 53.8 (18.7) | 21.2 (25.0) | 90.6 (0.8) | 95.7 (0.9) | 100.4 (23.5) | 84.2 (38.5) | 92.0 (10.7) | 109.4 (4.1) | - |
| **3.125** | 57.0 (12.8) | 4.2 (0.9) | 98.5 (4.1) | 90.3 (12.8) | 70.3 (69.8) | 45.6 (3.7) | 70.0 (0.5) | 98.1 (11.2) | - |
| **6.25** | 48.7 (12.3) | 4.0 (0.8) | 91.6 (2.3) | 96.8 (2.7) | 25.0 (27.6) | 32.2 (1.0) | 64.4 (1.7) | 100.7 (2.3) | - |
| **12.5** | 46.7 (3.5) | 1.6 (2.3) | 98.5 (3.0) | 95.5 (2.3) | 5.2 (1.1) | 14.4 (6.7) | 47.0 (4.4) | 77.0 (11.2) | - |
| **25** | 33.9 (1.3) | 1.4 (2.1) | 105.0 (12.2) | 97.0 (14.4) | 7.1 (1.9) | 7.7 (3.2) | 42.9 (3.5) | 77.9 (28.8) | - |
| **50** | 29.5 (3.8) | 1.4 (2.0) | 101.2 (2.3) | 100.8 (1.1) | 6.2 (0.8) | 7.7 (3.0) | 7.0 (3.9) | 60.1 (36.1) | - |
| **100** | 20.8 (8.1) | 1.5 (2.1) | 95.4 (7.1) | 99.7 (0.8) | 6.9 (0.3) | 7.8 (2.3) | 7.5 (4.4) | 19.9 (9.8) | - |
| **200** | 18.5 (8.5) | 1.7 (2.4) | 83.56 (4.9) | 91.3 (4.2) | 7.6 (0.6) | 8.6 (2.1) | 6.7 (2.3) | 19.0 (7.9) | 95.6 (13.3) |

RHG did not reduce cell viability at 200 mg/ml. These compounds were considered as not cytotoxic (Table 2). However, the other six compounds (Lemon, Peppermint, Triclosan, Fluoride, Cinnamon, and SDS) did cause a dose-dependent decrease in RHG viability, resulting in EC10 and EC50 values being obtained (Fig 2). Table 3 ranks the compounds in order of increasing irritant potency with the lowest EC50 value corresponding to the most irritant/cytotoxic compound, e.g. SDS is the most irritant whereas Xylitol, PEG, and SMP are classed as non-irritant. Notably, EC10 values did not follow the same order of ranking as the EC50 values as some compounds showed a very sharp dose-dependent increase in cytotoxicity whereas for others this was more gradual (slope differences in the line graphs of Fig 2). For example, Peppermint and Triclosan had lower EC10 values than expected when compared with the ranking of the EC50 values. Of note, when comparing the compound concentrations (mg/ml) used in toothpaste, for Lemon, Peppermint, Fluoride, Cinnamon, and SDS, the RHG EC50 values were within the range or lower than the corresponding concentration used in toothpaste. For Triclosan, the EC10 value corresponded to a lower concentration than that found in toothpaste.

Next, it was determined whether the compounds also had detrimental effects on RHG histology and whether these were in line with the EC10 and EC50 values (Fig 3). When the vehicle water was used this alone had a slightly detrimental effect on tissue histology when compared with unexposed RHG or AOO-exposed RHG and therefore needs to be taken into account when comparing effects caused by the compounds. Exposure of RHG to Xylitol, PEG, and SMP had no detrimental effect on tissue histology, in line with results obtained from the MTT assay. However, cell death was observed after exposure to lemon, peppermint, triclosan, fluoride, and SDS, which became more apparent as the compound concentration increased.

The vehicle for Triclosan, Peppermint, Cinnamon, and Lemon was AOO, for the others was water.

## Discussion

The MTT values exhibited for certain chemicals a significant variability at specific concentrations, particularly in the steep sections of the graphs, leading to high standard deviations. However, since the X-axis is on a logarithmic scale and is used to determine EC10 and EC50 values, this variability does not impact the final potency scoring. This study followed a previously

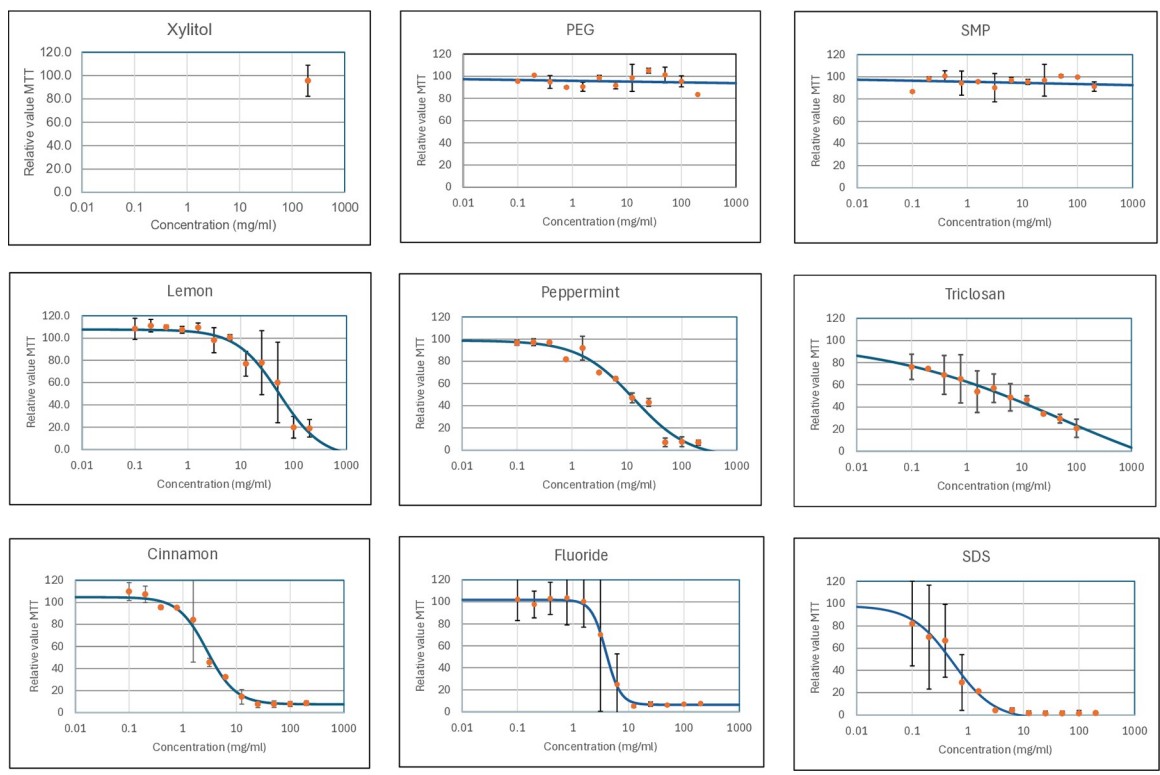

**Fig 2. Dose response effects of compounds on cell viability in in RHG model.** RHG was exposed to compounds as described in Materials and Methods. Cell viability was assessed by MTT reduction assay and is expressed relative to vehicle. Blue line = 90% viability cut-off and red line = 50% viability cut off. Results are expressed as average and standard deviation of two independent runs. The lines show the result of the 4-point logistic regression analysis used to calculate the EC10 and EC50 values (Table 2).

established standard operating procedure for the Epidermal Equivalent potency assay [29]. This procedure outlines how to perform dose-finding for any unknown coded chemical (with unknown molecular mass) to obtain an EC50 value by conducting two independent experiments. We previously demonstrated that this methodology is transferable to various commercially available and in-house reconstructed skin models, even in international ring studies [6].

**Table 3. Comparison of compound concentration used in toothpastes with EC50 and EC10 values obtained from RHG irritant potency assay.**

| Compound | Estimated amount in weight in toothpaste (w%) | Average product concentration in a toothpaste (mg/ml) | EC50 (mg/ml) | EC10 (mg/ml) | EC50 (mM) | EC10 (mM) |
|---|---|---|---|---|---|---|
| Xylitol | 10 | 159.4 | > 200 | > 200 | > 1314 | > 1314 |
| PEG | 0.5–2.0 | 7.97–31.87 | > 200 | > 200 | > 2629 | > 2629 |
| SMP | 8 | 127.49 | > 200 | > 200 | > 694 | > 694 |
| Lemon | 5 | 79.68 | 53.7 | 11.0 | 434.4 | 76.3 |
| Triclosan | 0.1–0.3 | 1.59–4.78 | 51.9 | 0.001 | 144.4 | 0.004 |
| Peppermint | 0.5–0.9 | 7.97–14.34 | 13.6 | 0.9 | 14.1 | 0.9 |
| Fluoride | 0.26 | 4.14 | 3.9 | 2.2 | 93.6 | 52.5 |
| Cinnamon | 1 | 15.94 | 2.8 | 1.0 | 20.7 | 7.8 |
| SDS | 1.2 | 19.12 | 0.5 | 0.1 | 1.9 | 0.2 |

EC50 value is compound concentration resulting in 50% loss of metabolic activity in RHG; EC10 value is compound concentration resulting in 10% loss of metabolic activity in RHG. Values were obtained by 4-point logistic regression analysis based on changes in metabolic activity (MTT).

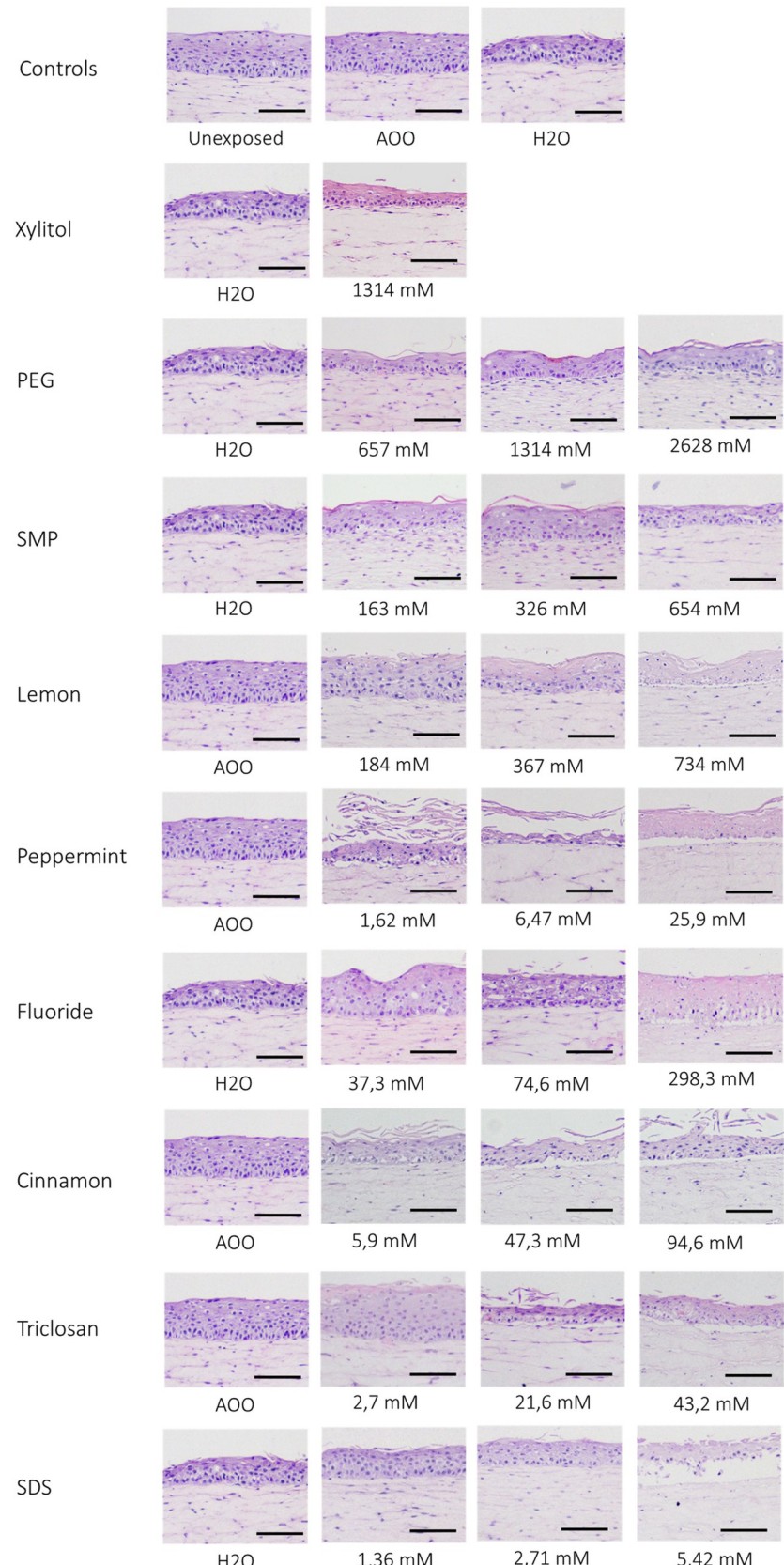

**Fig 3. Histological assessments of RHG.** RHG were unexposed, exposed to vehicle only, or to toothpaste ingredients, processed for paraffin embedment, and tissue sections (5 μm) were stained with Hematoxylin and Eosin stain for assessment of histology. The vehicle for Triclosan, Mint, Cinnamon and Lemon was acetone: Olive oil (AOO, 4:1), for the others it was water. Representative images are shown from three experiments, each performed with an intraexperiment duplicate. When the vehicle water was used this alone had a slightly detrimental effect on tissue histology when compared with unexposed RHG or AOO-exposed RHG and therefore needs to be taken into account when comparing effects caused by the compounds. Exposure of RHG to Xylitol, PEG, and SMP had no detrimental effect on tissue histology, in line with results obtained from the MTT assay. However, cell death was observed after exposure to lemon, peppermint, triclosan, fluoride, and SDS, which became more apparent as the compound concentration increased.

Now, we show that it is not only suitable for assessing skin irritant potency in reconstructed human skin models but can also be applied to reconstructed human gingival models.

Overall, our findings highlight the importance of carefully considering the ingredients used in toothpastes and the potential effects they may have on human health. Our results show that a number of ingredients commonly found in toothpastes may have negative effects on the human gingiva when they are applied in toothpaste or mouth rinses. While Xylitol, PEG, and SMP did not appear to affect cell viability, the common ingredients Lemon, Peppermint, Cinnamon Triclosan, Fluoride and SDS do affect cell viability in our RHG test, indicating that toothpastes containing these ingredients could have a negative impact on the oral mucosa and eventually, when swallowed also on the gut mucosa.

Even though it is not possible to directly compare concentrations present in toothpaste with in vitro RHG obtained EC10 and EC50 values, it should be noted that especially in the case of SDS, the concentration present in toothpaste formulations significantly surpasses our experimental levels by more than 40-fold. This stark contrast strongly suggests that SDS is likely to induce a marked adverse effect due to its nature as an irritant surfactant that dissolves lipids. It is important to emphasize that SDS is not a sensitizer, and the potential for an allergic response is not a concern. Furthermore, given its relatively high concentration in toothpaste, SDS could potentially disrupt the proteins responsible for the internal immunological defense of saliva and compromise the protective function of the oral mucosa.

In our experimental setup, the mucosa is not protected by a mucous layer as it is in the mouth. In particular, SDS as a surface-active agent can degrade this mucus layer which therefore makes it more likely that all toxic components of toothpastes containing SDS can reach and penetrate the living gingival cell layers. In this aspect, it can be expected that SDS- free toothpastes are considerably less irritant than SDS-containing toothpastes.

The MTT results obtained from RHG exposed to Fluoride are remarkable because at lower concentrations they lead to no effect on cell viability but in a relatively small concentration increase immediately lead to large effects on cell viability (small difference between EC10 and EC50). It is worth noting that both Fluoride and SDS, when swallowed, can pass through the stomach without degradation and potentially affect the intestinal mucosa [3,30] (Alépée et al. 2014; Liu et al. 2019). It would be beneficial to further investigate the effects of these ingredients on the intestinal mucosa.

Although flavors are generally added in low concentrations, it is important to note that Lemon, Peppermint, and Cinnamon, in addition to having irritant properties, are known as sensitizers and can trigger an allergic reaction even at low concentrations. In our observations, it's worth noting that Triclosan has faced considerable restrictions in toothpaste formulations in the Netherlands, probably stemming from safety concerns.

Specific data regarding applied concentrations of these ingredients in toothpaste were not provided by the manufacturers (Table 2). The next step is to proceed with this study, not only with the single ingredients but with complete toothpaste (mixtures) that differ in ingredients

in an attempt to determine which ingredients may have the worst side effects and determine as to whether additional cytotoxicity could be caused by the ingredient mixtures. Mixtures may require different concentrations to cause a toxic effect on cell viability as individual ingredients can either synergistically enhance or mutually diminish their impact on cell viability. For this reason, we cannot reliably compare these results with clinical effects of toothpaste.

Consequently, the MTT assay results primarily offer a comparative insight into toxicity levels.

The findings of this study could contribute to the development of safer toothpaste formulations by identifying toothpaste ingredients that cause toxicity to oral fibroblasts and epithelial cells. Overall, this study highlights the importance of evaluating toothpaste ingredients for their toxicity to ensure their safety for oral use.

## Conclusion

While Xylitol, PEG, and SMP did not appear to affect cell viability, the concentrations of Lemon, Peppermint, Cinnamon, and SDS might harm the oral mucosa. Triclosan and Fluoride concentrations exceeded the EC10 value but stayed below the EC50 value.

## Supporting information

**S1 File. MTT data-file contains all raw data and calculations.**
(XLSX)

## Author Contributions

**Conceptualization:** Shaira R. Kasi, Sanne Roffel, Mutlu Özcan, Susan Gibbs, Albert J. Feilzer.

**Data curation:** Shaira R. Kasi, Sanne Roffel, Albert J. Feilzer.

**Formal analysis:** Sanne Roffel, Susan Gibbs, Albert J. Feilzer.

**Investigation:** Shaira R. Kasi, Sanne Roffel, Mutlu Özcan, Susan Gibbs, Albert J. Feilzer.

**Supervision:** Susan Gibbs, Albert J. Feilzer.

**Writing – original draft:** Shaira R. Kasi, Sanne Roffel.

**Writing – review & editing:** Mutlu Özcan, Susan Gibbs, Albert J. Feilzer.

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
