## [Decision Letter · Decision Letter 0]

11 Sep 2024

PONE-D-24-31912In vitro cytotoxicity (irritant potency) of toothpaste ingredientsPLOS ONE

Dear Dr.  Feilzer,

Thank you for submitting your manuscript to PLOS ONE. After careful consideration, we feel that it has merit but does not fully meet PLOS ONE’s publication criteria as it currently stands. Therefore, we invite you to submit a revised version of the manuscript that addresses the points raised during the review process.

We look forward to receiving your revised manuscript.

Kind regards,

Tanay Chaubal

Academic Editor

PLOS ONE

Journal Requirements:

1. Please ensure that your manuscript meets PLOS ONE's style requirements, including those for file naming. The PLOS ONE style templates can be found at https://journals.plos.org/plosone/s/file?id=wjVg/PLOSOne_formatting_sample_main_body.pdf andhttps://journals.plos.org/plosone/s/file?id=ba62/PLOSOne_formatting_sample_title_authors_affiliations.pdf

2.  We note that your Data Availability Statement is currently as follows: “All relevant data are within the manuscript and its Supporting Information files.”

Please confirm at this time whether or not your submission contains all raw data required to replicate the results of your study. Authors must share the “minimal data set” for their submission. PLOS defines the minimal data set to consist of the data required to replicate all study findings reported in the article, as well as related metadata and methods (https://journals.plos.org/plosone/s/data-availability#loc-minimal-data-set-definition). For example, authors should submit the following data: - The values behind the means, standard deviations and other measures reported; - The values used to build graphs; - The points extracted from images for analysis. Authors do not need to submit their entire data set if only a portion of the data was used in the reported study. If your submission does not contain these data, please either upload them as Supporting Information files or deposit them to a stable, public repository and provide us with the relevant URLs, DOIs, or accession numbers. For a list of recommended repositories, please see https://journals.plos.org/plosone/s/recommended-repositories. If there are ethical or legal restrictions on sharing a de-identified data set, please explain them in detail (e.g., data contain potentially sensitive information, data are owned by a third-party organization, etc.) and who has imposed them (e.g., an ethics committee). Please also provide contact information for a data access committee, ethics committee, or other institutional body to which data requests may be sent. If data are owned by a third party, please indicate how others may request data access.

Additional Editor Comments:

** I thank you for submitting an interesting manuscript. However, there are comments made by reviewers which potentially could make your manuscript more robust.**

Reviewers' comments:

Reviewer's Responses to Questions

**Comments to the Author**

1. Is the manuscript technically sound, and do the data support the conclusions?

Reviewer #1: Yes

Reviewer #2: Yes

2. Has the statistical analysis been performed appropriately and rigorously? 

Reviewer #1: N/A

Reviewer #2: N/A

3. Have the authors made all data underlying the findings in their manuscript fully available?

Reviewer #1: No

Reviewer #2: Yes

4. Is the manuscript presented in an intelligible fashion and written in standard English?

Reviewer #1: Yes

Reviewer #2: Yes

5. Review Comments to the Author

Reviewer #1: The study highlights the potential for certain toothpaste ingredients to harm the cells. The strength of the paper is the use of a bilayered RHG model, which enhances the relevance of the findings to human oral health. However, the study's limitation lies in the inability to directly compare in vitro concentrations with those in commercial toothpaste, which could affect the interpretation of the results. These are stated as limitation in the manuscript.

Authors are requested to make certain clarifications in the paper.

1. Were the experiments conducted in triplicates/duplicates? Were the data from MTT values normalized? before comparing? If not, it is advised to show the data (MTT observations) properly in a table form.

2. What statistcal analyses was performed? please state and show statistical values legibly.

3. The changes in the histology is not mentioned in detail anywhere in the manuscript. It is required to state and show observde changes clearly. what are the changes observed.

4. Photographs depicting the RHG model and preparation will benefit readers. it is recommended to add them.

5. The discussion part can be improved in terms of comparing the obsrvations to exisiting evidences in the literature rather than just reporting the obervations.

Reviewer #2: The aim and purpose of the work is very well exposed. This study reminds so interesting and have got and excellent clinical transpolation. Please describe in more detail the statistical analysis used.

6. PLOS authors have the option to publish the peer review history of their article (what does this mean?). If published, this will include your full peer review and any attached files.

Reviewer #1: **Yes: **Spoorthi Ravi Banavar

Reviewer #2: **Yes: **Evelin Bachmeier

---

## [Author Response · Author response to Decision Letter 0]

29 Oct 2024

Reviewers' comments:

Reviewer's Responses to Questions

Comments to the Author

1. Is the manuscript technically sound, and do the data support the conclusions?

Reviewer #1: Yes

Reviewer #2: Yes

2. Has the statistical analysis been performed appropriately and rigorously? 

Reviewer #1: N/A

Reviewer #2: N/A

3. Have the authors made all data underlying the findings in their manuscript fully available?

Reviewer #1: No

 Original raw data are uploaded as supplementary file 

Reviewer #2: Yes

4. Is the manuscript presented in an intelligible fashion and written in standard English?

Reviewer #1: Yes

Reviewer #2: Yes

5. Review Comments to the Author

Reviewer #1: The study highlights the potential for certain toothpaste ingredients to harm the cells. The strength of the paper is the use of a bilayered RHG model, which enhances the relevance of the findings to human oral health. However, the study's limitation lies in the inability to directly compare in vitro concentrations with those in commercial toothpaste, which could affect the interpretation of the results. These are stated as limitation in the manuscript.

Authors are requested to make certain clarifications in the paper.

1. Were the experiments conducted in triplicates/duplicates? Were the data from MTT values normalized? before comparing? If not, it is advised to show the data (MTT observations) properly in a table form.

The experiments are conducted in duplicate.

The data from the MTT are per culture and test material compared to that of unexposed RHG of the same culture as well as to the values of exposure to the test vehicle only. The data are ‘normalized’ to the results of the exposure to the test vehicle per culture.

Clarification is added to the text: 

“All experiments were conducted in two independent experiments, each using a different batch of RHG which had been grown separately at different times.”

“The data from the MTT assay are per RHG exposed to a test material compared to that of unexposed RHG of the same RHG batch as well as to the values obtained from exposure to the test vehicle only. The data are shown normalized to the results of the exposure to the test vehicle per culture.”

2. What statistical analyses was performed? please state and show statistical values legibly. 

Data in duplicate are reported as average and standard deviation. The calculation of the EC10 and EC 50 values was done by 4-point logistic regression.

The text is changed: “Values were obtained by linear regression 4 point logistic analysis based on changes in metabolic activity (MTT). The web-based EC50 Calculator of AAT Bioquest [28] was used.”

3. The changes in the histology is not mentioned in detail anywhere in the manuscript. It is required to state and show observed changes clearly. What are the changes observed.

The observed changes are added to the figure legend (Fig. 3)

4. Photographs depicting the RHG model and preparation will benefit readers. it is recommended to add them.

A figure (Fig. 1) depicting the RHG model in a MTT is added to the manuscript.

5. The discussion part can be improved in terms of comparing the observations to existing evidences in the literature rather than just reporting the observations.

The plan involved initially assessing the toxicity of all potentially harmful ingredients in toothpaste using the MTT assay based on the RHG model. After this study, we will begin testing toothpastes containing these ingredients. The current results reflect a single exposure to the test materials. For instance, when testing a toothpaste, these substances may either attenuate or potentiate each other’s effects. Additionally, we emphasized that our MTT assay was conducted in an RHG model rather than with single cell types. Therefore, we decided not to compare our observations with the existing scientific literature at this stage. Furthermore, we have already demonstrated that the RHG model produces similar tissue responses when exposed to materials, as shown in Buskermolen’s study.

Reviewer #2: The aim and purpose of the work is very well exposed. This study reminds so interesting and have got and excellent clinical transpolation. Please describe in more detail the statistical analysis used.

After the thorough review of the PLOS reviewers and during answering their questions, we realized that the linear regression used on the MTT data was outdated. Consequently, we reanalyzed the data using a 4-point logistic analysis. We replaced the bar graphs ( Fig. 2) with logistic regression plots, which offer better insights into the slope differences between the test materials.

We have clarified in the text that the 4-point logistic analysis was performed using ATT’s web-based calculator. All necessary data to reproduce the analysis have been included in a supplementary file.

6. PLOS authors have the option to publish the peer review history of their article (what does this mean?). If published, this will include your full peer review and any attached files.

Do you want your identity to be public for this peer review? For information about this choice, including consent withdrawal, please see our Privacy Policy.

Reviewer #1: Yes: Spoorthi Ravi Banavar

Reviewer #2: Yes: Evelin Bachmeier

---

## [Decision Letter · Decision Letter 1]

27 Dec 2024

PONE-D-24-31912R1In vitro cytotoxicity (irritant potency) of toothpaste ingredientsPLOS ONE

Dear Dr. Feilzer,

Thank you for submitting your manuscript to PLOS ONE. After careful consideration, we feel that it has merit but does not fully meet PLOS ONE’s publication criteria as it currently stands. Therefore, we invite you to submit a revised version of the manuscript that addresses the points raised during the review process.

**ACADEMIC EDITOR: I thank you for responding to the reviewers' comments. However, there still minor revisions which are required as per reviewers' comments.**

We look forward to receiving your revised manuscript.

Kind regards,

Tanay Chaubal

Academic Editor

PLOS ONE

Journal Requirements:

Reviewers' comments:

Reviewer's Responses to Questions

**Comments to the Author**

1. If the authors have adequately addressed your comments raised in a previous round of review and you feel that this manuscript is now acceptable for publication, you may indicate that here to bypass the “Comments to the Author” section, enter your conflict of interest statement in the “Confidential to Editor” section, and submit your "Accept" recommendation.

Reviewer #1: All comments have been addressed

Reviewer #2: All comments have been addressed

2. Is the manuscript technically sound, and do the data support the conclusions?

Reviewer #1: Yes

Reviewer #2: Yes

3. Has the statistical analysis been performed appropriately and rigorously? 

Reviewer #1: Yes

Reviewer #2: Yes

4. Have the authors made all data underlying the findings in their manuscript fully available?

Reviewer #1: Yes

Reviewer #2: Yes

5. Is the manuscript presented in an intelligible fashion and written in standard English?

Reviewer #1: Yes

Reviewer #2: Yes

6. Review Comments to the Author

Reviewer #1: Thank you for addressing our comments. The manuscript describes the methods and results as required. However, it is recommended to always perform experiments in triplicates and not duplicates. Additionally, I would like to point that the standard deviations noted in the experiments are very wide and this is a point of concern especially when done in duplicates. If this cannot be addressed now, please mention in the limitation. Additionally, it is recommended to depict the pictures of the RHG model before and during histology. this will benefit readers.

Reviewer #2: All request have been answered.

Recommendatons have been attended. The article is very interesting. Ethical aspects have been taken into account.

7. PLOS authors have the option to publish the peer review history of their article (what does this mean?). If published, this will include your full peer review and any attached files.

Reviewer #1: **Yes: **Spoorthi Ravi Banavar

Reviewer #2: **Yes: **Evelin Bachmeier

---

## [Author Response · Author response to Decision Letter 1]

9 Jan 2025

Reviewer #1: Thank you for addressing our comments. The manuscript describes the methods and results as required. However, it is recommended to always perform experiments in triplicates and not duplicates. Additionally, I would like to point that the standard deviations noted in the experiments are very wide and this is a point of concern especially when done in duplicates. If this cannot be addressed now, please mention in the limitation. Additionally, it is recommended to depict the pictures of the RHG model before and during histology. this will benefit readers.

We add the following text to the discussion section:

The MTT values exhibited for certain chemicals a significant variability at specific concentrations, particularly in the steep sections of the graphs, leading to high standard deviations. However, since the X-axis is on a logarithmic scale and is used to determine EC10 and EC50 values, this variability does not impact the final potency scoring. This study followed a previously established standard operating procedure for the Epidermal Equivalent potency assay.[29] This procedure outlines how to perform dose-finding for any unknown coded chemical (with unknown molecular mass) to obtain an EC50 value by conducting two independent experiments. We previously demonstrated that this methodology is transferable to various commercially available and in-house reconstructed skin models, even in international ring studies. [6] Now, we show that it is not only suitable for assessing skin irritant potency in reconstructed human skin models but can also be applied to reconstructed human gingival models. 

N.B. also we add one reference M. A. T. Teunis et al., “International ring trial of the epidermal equivalent sensitizer potency assay: Reproducibility and predictive capacity,” ALTEX, vol. 31, no. 3, pp. 251–268, 2014, doi: 10.14573/altex.1308021.

Additionally we changed figure 3 in accordance with the suggestion of the reviewer.

---

## [Decision Letter · Decision Letter 2]

19 Jan 2025

In vitro cytotoxicity (irritant potency) of toothpaste ingredients

PONE-D-24-31912R2

Dear Dr. Albert Joseph Feilzer, 

We’re pleased to inform you that your manuscript has been judged scientifically suitable for publication and will be formally accepted for publication once it meets all outstanding technical requirements.

Kind regards,

Tanay Chaubal

Academic Editor

PLOS ONE

Additional Editor Comments (optional):

Reviewers' comments:

Reviewer's Responses to Questions

**Comments to the Author**

1. If the authors have adequately addressed your comments raised in a previous round of review and you feel that this manuscript is now acceptable for publication, you may indicate that here to bypass the “Comments to the Author” section, enter your conflict of interest statement in the “Confidential to Editor” section, and submit your "Accept" recommendation.

Reviewer #1: All comments have been addressed

2. Is the manuscript technically sound, and do the data support the conclusions?

Reviewer #1: Yes

3. Has the statistical analysis been performed appropriately and rigorously? 

Reviewer #1: Yes

4. Have the authors made all data underlying the findings in their manuscript fully available?

Reviewer #1: Yes

5. Is the manuscript presented in an intelligible fashion and written in standard English?

Reviewer #1: Yes

6. Review Comments to the Author

Reviewer #1: Thank you for addressing my review comments. If possible, it would be good (recommend) including detailed image of RHG model and its preparation, before and after histology.

7. PLOS authors have the option to publish the peer review history of their article (what does this mean?). If published, this will include your full peer review and any attached files.

Reviewer #1: **Yes: **Spoorthi Ravi Banavar

---

## [Editor Report · Acceptance letter]

22 Jan 2025

PONE-D-24-31912R2 

PLOS ONE

Dear Dr. Feilzer, 

I'm pleased to inform you that your manuscript has been deemed suitable for publication in PLOS ONE. Congratulations! Your manuscript is now being handed over to our production team.

Kind regards, 

on behalf of

Dr. Tanay Chaubal 

Academic Editor

PLOS ONE